# Mitochondrial Dysfunction in Lung Resident Mesenchymal Stem Cells from Idiopathic Pulmonary Fibrosis Patients

**DOI:** 10.3390/cells12162084

**Published:** 2023-08-17

**Authors:** Josep Mercader-Barceló, Aina Martín-Medina, Joan Truyols-Vives, Gabriel Escarrer-Garau, Linda Elowsson, Ana Montes-Worboys, Carlos Río-Bocos, Josep Muncunill-Farreny, Julio Velasco-Roca, Anna Cederberg, Måns Kadefors, Maria Molina-Molina, Gunilla Westergren-Thorsson, Ernest Sala-Llinàs

**Affiliations:** 1iRESPIRE Research Group, Health Research Institute of the Balearic Islands (IdISBa), 07120 Palma, Spain; 2MolONE Research Group, University of the Balearic Islands, 07122 Palma, Spain; 3Lung Biology, Department of Experimental Medical Science, Lund University, 08908 Lund, Sweden; 4ILD Unit, Respiratory Department, University Hospital of Bellvitge-Bellvitge Biomedical Research Institute (IDIBELL), Hospitalet de Llobregat, 08908 Barcelona, Spain; 5Health Research Institute of the Balearic Islands (IdISBa), 07120 Palma, Spain; 6Centre of Biomedical Research Network in Respiratory Diseases (CIBERES), 28029 Madrid, Spain; 7Respiratory Department, Son Espases University Hospital, 07120 Palma, Spain

**Keywords:** idiopathic pulmonary fibrosis, lung resident mesenchymal stem cell, mitochondria, oxidative phosphorylation, mitophagy, apoptosis, transforming growth factor β

## Abstract

Idiopathic pulmonary fibrosis (IPF) is characterized by an aberrant repair response with uncontrolled turnover of extracellular matrix involving mesenchymal cell phenotypes, where lung resident mesenchymal stem cells (LRMSC) have been supposed to have an important role. However, the contribution of LRMSC in lung fibrosis is not fully understood, and the role of LRMSC in IPF remains to be elucidated. Here, we performed transcriptomic and functional analyses on LRMSC isolated from IPF and control patients (CON). Both over-representation and gene set enrichment analyses indicated that oxidative phosphorylation is the major dysregulated pathway in IPF LRMSC. The most relevant differences in biological processes included complement activation, mesenchyme development, and aerobic electron transport chain. Compared to CON LRMSC, IPF cells displayed impaired mitochondrial respiration, lower expression of genes involved in mitochondrial dynamics, and dysmorphic mitochondria. These changes were linked to an impaired autophagic response and a lower mRNA expression of pro-apoptotic genes. In addition, IPF TGFβ-exposed LRMSC presented different expression profiles of mitochondrial-related genes compared to CON TGFβ-treated cells, suggesting that TGFβ reinforces mitochondrial dysfunction. In conclusion, these results suggest that mitochondrial dysfunction is a major event in LRMSC and that their occurrence might limit LRMSC function, thereby contributing to IPF development.

## 1. Introduction

Idiopathic pulmonary fibrosis (IPF) is an accelerated aging-related disease of poor prognosis with no curative treatment. Pulmonary function in IPF patients progressively declines concomitant to the accumulation of extracellular matrix (ECM) proteins and the loss of alveolar epithelial cell (AEC) identity [1,2]. Such events occur because of an aberrant response to epithelial damage mediated and sustained, in part, by the transforming growth factor β (TGFβ) cytokine [3]. In established IPF, ECM proteins are overproduced by mesenchymal cells such as fibroblasts in response to TGFβ secreted by the damaged AEC [4]. As the damage is unresolved, TGFβ is permanently secreted, thereby inducing fibroblast proliferation and promoting the differentiation of fibroblasts into myofibroblasts, the occurrence of which also contributes to the loss of normal lung architecture and ECM protein production [2,4]. Myofibroblasts can also emerge from other cell types, such as pericytes and AEC, through the so-called epithelial to mesenchymal transition (EMT) [5].

The development of IPF is influenced by extrinsic and intrinsic risk factors, but those that lead to the onset of this disease are essentially unknown. Environmental agents such as cigarette smoking, airborne pollutants, and airway microbiota are generally accepted as risk factors [1,2], whereas the role of gut microbiota and diet has been recently posed [6]. Intrinsic factors include the presence of comorbidities, such as gastroesophageal reflux, pulmonary hypertension, and diabetes [1,7], and the carriage of gene variants and mutations associated with increased mucous accumulation, telomere shortening, increased endoplasmic reticulum stress, and apoptotic signaling [8]. Age is a crucial risk factor for developing IPF. Indeed, the hallmarks of aging, such as telomere shortening [9], loss of proteostasis, cellular senescence [10,11], and mitochondrial dysfunction [12,13], have been described in AEC, lung fibroblasts, and immune cells, the presence of which drives IPF. On the other hand, the molecular characterization of lung resident mesenchymal stem cells (LRMSC) has been long forgotten, despite their pivotal role in tissue repair. Just recently, the role of lung resident precursor cells in the aberrant repair of IPF has started to be investigated. Our hypothesis is that IPF LRMSC exhibit molecular alterations that are translated into an altered functionality and contribute to IPF development.

Both LRMSC and lung resident mesenchymal progenitor cells (LRMPC) are stromal cells that proliferate and differentiate to act in the repair and regeneration of lung tissue. LRMSC are multipotent cells and LRMPC are committed to differentiating into fibroblasts. It has been shown that LRMPC are the cells of origin for fibrosis-mediating myofibroblasts [14] that acquire cell-autonomous fibrogenicity early in their differentiation trajectory [15]. Although evidence has suggested that LRMPC actively contribute to IPF, evidence in animal models suggests that LRMSC could be involved in pulmonary fibrosis as a consequence of MSC exhaustion [16] and thereby limiting the regeneration of alveolar epithelium, and/or because they actively contribute to the dysregulated repair mechanism by differentiating into myofibroblasts [17]. Other studies conducted on LRMSC from lung damage animal models showed that LRMSC are able to differentiate into myofibroblasts through the activation of hedgehog and Wnt/B-catenin signaling [18,19] and suggested that they could be a major source of myofibroblasts, thereby actively contributing to pulmonary fibrosis progression [20,21,22,23,24]. In a study with human LRMSC involving IPF patients [25], the authors showed that IPF LRMSC displayed a higher expression of genes involved in inflammation, oxidative stress, hypoxia, and ECM pathway than control LRMSC and, in co-culture with control MSC or fibroblasts, induced a pathological phenotype on the surrounding cells. In the present study, we conducted a transcriptomic and functional characterization of LRMSC from different cohorts by comparing them to LRMSC obtained from individuals without interstitial lung disease. IPF LRMSC present significant differences in pathways and biological processes, which include downregulation in oxidative phosphorylation. IPF LRMSC present signs of mitochondrial dysfunction that could play a role in IPF development.

## 2. Materials and Methods

### 2.1. Patient Recruitment

The collection of human LRMSC was approved by the Balearic Islands (IB 1991/13 PI; IB 3335/16 PI) and Swedish (2008/413 (2022-01221-02), 657-12 (13 Setember 2012, 2006/91) Ethical Committees. LRMSC were obtained from the reservoirs of three independent cohorts—University Hospital of Bellvitge (Barcelona, Spain), University Hospital of Son Espases (HUSE, Palma de Mallorca, Spain), and Lund University Hospital (Lund, Sweeden)—and designated either to CON or IPF groups. IPF patients were included in this study according to the guidelines followed for IPF diagnosis based on ATS and ERS criteria [26]. In total, samples from 11 IPF patients and 14 control individuals were used in this study (Table 1). The number of patient samples used in each analysis is detailed in the figure legends.

### 2.2. LRMSC Isolation and Validation

Lung tissue from IPF patients was obtained from biopsies or explanted tissue from IPF patients undergoing transplantation (Lund and Bellvitge cohorts) performed for diagnosis purposes, whereas lung tissue from control individuals was obtained from lungs of postmortem persons (HUSE cohort) or from the surrounding tissue of tumor resections (Lund cohort).

All tissue samples were processed to isolate LRMSC using standardized protocols. For the Bellvitge and HUSE cohorts, tissues were minced and digested with collagenase for 30 min at 37 °C. The sample was diluted with PBS and filtered through 100 µm and 75 µm cell filters and washed 3 times in PBS. Cells from the resulting pellet were cultured for 4–6 days in αMEM medium (Biowest, L0475-500, Nuaillé, France) supplemented with 10% FCS (Biowest, S1810-500) and 1% penicillin/streptomycin (Capricorn Scientific GmbH, Ebsdorfergrund, Germany) inside a 37 °C, 5% CO_2_, 98% humidity incubator. Non-adhered cells were removed by changing the media, and the adhered LRMSC were maintained in culture for experiments. The mesenchymality of the cells was confirmed using the BD Stemflow^TM^ human MSC analysis kit (BD Biosciences, 562245, New Jersey, USA), the human mesenchymal stem cell functional identification kit (R&D, SC006), and by fitting the results of the gene expression microarray (described below) into the silico Rohart MSC test, which accurately discriminates MSC from fibroblasts and other cell types and is available at https://www.stemformatics.org/ [27]. For the Lund cohort, LRMSC were isolated from human lung tissue explants according to a previously published protocol [28] that has been demonstrated to generate mesenchymal cell cultures from human lung tissue that fulfill the ISCT minimal criteria to define MSC [29], including immunophenotyping and tri-lineage differentiation. Briefly, distal lung tissue was dissected from lung explants and parenchymal tissue was enriched by removing the pleura, larger airways, and vessels. The tissue was washed with PBS, cut into small pieces (<3 mm^3^) and enzymatically dissociated with 300 U/mL collagenase type I (Gibco, Paisley, Scotland), 1 mg/mL hyaluronidase (Serva Electrophoresis GmbH, Heidelberg, Germany), and 100 U/mL DNase I (Sigma-Aldrich, Saint Louis, MO, USA) in PBS at 37 °C for 1.5–2 h. Tissue pieces were subsequently separated from the cell suspension by filtering through a 100 mm cell strainer. For some patients/donors, cells were treated with a buffer containing 155 mM NH_4_CL and 10 mM KHCO_3_ to lyse red blood cells. Finally, cells were seeded on tissue culture plastic in StemMACS MSC Expansion Media (Miltenyi Biotec, 130-091-680, Bergisch Gladbach, Germany) supplemented with 1% antibiotic/antimycotic solution (Sigma-Aldrich) and cultured at 37 °C and 5% CO_2_. Cells were passaged once colonies had formed after approximately 14 days.

### 2.3. Transcriptomic Analysis

Four samples of each experimental group from the Bellvitge and HUSE cohorts were included in this analysis. Differential gene expression analysis between IPF and CON LRMSC was approached using the GeneChip^®^ Gene 1.0 ST Array System (Affymetrix, Santa Clara, CA, USA), containing 36,079 probe sets. Genes with an absolute fold change >2 and adjusted *p*-value for the false discovery rate (FDR) <0.05 were considered differentially expressed. Gene expression was calculated using the limma package from Bioconductor (R version 4.2.2 software). Microarray information is available on the Gene Expression Omnibus data repository.

The identification of altered pathways in IPF cells was approached by performing both over-represented analysis (ORA) and gene-set enrichment analysis (GSEA). ORA was made from a list of genes that were expressed significantly differently, whereas GSEA was made from all the genes included in the array ranked by fold change. Pathways were identified using the Kyoto Encyclopedia of genes and genomes (KEGG), applying an adjusted *p*-value for BH <0.05. Biological processes (BP) were identified by gene ontology (GO) analysis by applying an adjusted *p*-value for B&H <0.05. Enriched GO terms were grouped by semantic analysis using the rrvgo package from Bioconductor using the REVIGO clustering algorithm [30]. After GSEA, representative GO terms were grouped by applying a 0.8 correlation factor.

### 2.4. mRNA Analysis by qPCR

RNA was isolated from cultured LRMSC using the Extractme Total RNA Kit (Blirt, EM09.1, Gdansk, Poland) following the manufacturer’s protocol. In total, 150 ng of RNA were subjected to cDNA synthesis by following the instructions of the Transcriptme RNA kit (Blirt, RT31, Gdansk Poland). Finally, cDNA was used for PCR analysis using the SensiIFASTT^M^ SYBR^®^ No-ROX Kit (Bioline, BIO-98005, Antipolo City, Philippines) and validated human primers. *β2 microglobulin* (*B2M*) was used as a housekeeping gene. Forward and reverse primer sequences were ACGCCCACTGAAAAAGATGAG and ATCTTCAAACCTCCATGATGC for *B2M*, GTACGAGCTCATGAAAGTGTTG and ACATAGTGCTTCTGAGAGATGA for *cytochrome c oxidase subunit IV* (*COX-IV*), TATTGCTAGGGTGGCGCTTC and ACCACAAGGCACACCTACAC for *ATP synthase subunit VI* (*ATP6*), GGAGACGTGACCACTGACAATGA and TGTTGGCTGCCAGTAAGAG for *peroxisome proliferator-activated receptor gamma coactivator 1 alpha* (*PGC1A*), CAAGGATGTTGTCGGATTTCG and CGTTATGAAGAACTATCCCTG for *PTEN-induced kinase 1* (*PINK1*), GAACTGGACCCCGTTACCAC and TTGATCACGGTGCTCTTCCC for *mitofusine 2* (*MFN2*), and AGGCAGGAGGAGGACAGTTA and GTGCTCTGCACACGATGTAG for *phosphoglycerate mutase family member 5 long isoform* (*PGAM5L*).

### 2.5. Mitochondrial Respiration Assay

Mitochondrial respiration was measured in LRMSC using the Seahorse XF Cell Mito Stress Test Kit (Agilent, Santa Clara, CA, USA) following the user guide. Prior to assay, cells were seeded in octuplicates per condition in the Seahorse XF Cell Culture Microplate of 96 wells at 5000 cells/w using complete medium (10% FCS, 1% P/S, αMEM), and the cartridge was hydrated in calibrant solution and kept in a non-CO_2_ incubator. The next day, cells were put in starvation medium (2% FCS, 1% P/S, αMEM). After 24 h, cell medium was replaced by assay medium (Seahorse XF DMEM, #103680-100) supplemented with 1 mM pyruvate, 2 mM glutamine, and 10 mM glucose and the cartridge was loaded with the kit’s compounds prepared at the following concentrations: oligomycin (0.5 µM), FCCP (1 µM), and rotenone/AA (0.5 µM). The cell plate and cartridge were then loaded into the Seahorse XFe96 Analyzer instrument and the assay was run. Measurements were taken using the default settings and the following parameters were saved for statistics: oxidative respiration kinetic curve for the whole assay, basal and maximal respiration, and spare capacity.

### 2.6. Autophagy Assay

Autophagy vacuoles were labelled in living cells using an autophagy assay kit (Sigma-Aldrich, #MAK138, Saint Louis, MO, USA) following the manufacturer’s protocol. Briefly, cells were first seeded in black 96-well plates with a clear bottom at 5000 cells/well in complete medium (10% FCS, 1% P/S, αMEM). The next day, medium was replaced by autophagosome detection reagent solution and cells were incubated for 45 min. The cells were then washed 3 times with wash buffer and the fluorescence was intensity measured at 360/520 nm using a microplate reader (Synergy H1). Right afterwards, the fluorescence was visualized by microscope (Cell Observer, Zeiss).

### 2.7. Electromicroscopy

LRMSC from CON (*n* = 5) and IPF tissues (*n* = 4) from the Lund cohort were cultured until confluence. The cells were trypsinized and counted. In total, 15,000 cells were centrifuged, the supernatant was removed, and 1 mL of 2% paraformaldehyde and 2% glutaraldehyde in 0.1 M Sörensen’s phosphate buffer were added and incubated at room temperature for 1 h. The pellet was washed three times in Sörensen’s buffer and postfixed in 2% osmium tetroxide, dehydrated in increasing concentrations of acetone, and finally embedded in Polybed 812. The samples were sectioned with a diamond knife on Leica EM UC7. Sections were mounted on pioloform-coated maxtaform H5 grid and stained with 4% uranylacetat and 1% lead citrate. The samples were examined with a Tecnai BioTwin 120 kV microscope.

### 2.8. TGFβ Treatments

To analyze the LRMSC response to TGFβ, the cells were plated in 12-well plates at 100,000 cells/well in complete medium and starved the next day by decreasing the FCS concentration to 2%. After 6–24 h of starvation, cells were treated in duplicate at the indicated TGFβ (R&D Systems, 240-B, Minneapolis, MN, USA) concentration and time. Control cells received 0.1% BSA.

### 2.9. Mitochondrial DNA Quantification

Mitochondrial DNA levels were measured by real-time PCR using primers for the *MT-ATP6* and the *RPP30* nuclear gene for result normalization. LRMSC were seeded in normal medium in a 24-well plate at 50,000 cells per well. DNA was extracted with the Dneasy Blood & Tissue Kit following the kit’s instructions (Qiagen, Germantown, Maryland, USA). Target sequences were amplified using the SensiIFASTT^M^ SYBR^®^ No-ROX Kit with the following primers: *MT-ATP6* (fw: 5′ ACAACTAACCTCCTCGGACT3′, rv: 5′TGCCTTGTGGTAAGAAGTAGTGG3′) and *RPP30* (fw: 5′ GATTTGGACCTGCGAGCG 3′, rv: 5′ GCGGCTGTCTCCACAAGT).

### 2.10. Mitochondrial Membrane Potential (MMP) Assessment

Mitochondrial membrane potential was measured using tetramethylrhodamine methyl ester (TMRM, Thermo Fisher Scientific, Eugene, OR, USA). Carbonyl cyanide 3-chlorophenylhydrazone (CCCP, Sigma-Aldrich, Saint Louis, MO, USA) was used as a negative control and wells without TMRM or CCCP were included as blanks. Additional staining with Hoescht was used to normalize the values to the cell number. LRMSC were seeded in black 96-well plates with a clear bottom (SPL, 33,396) at 10,000 cells/well. The next day, the cells were washed once with PBS and incubated in 2% FCS, 1% P/S, and αMEM medium containing the drugs. At least 4 replicates were used per condition. After 30 min of incubation time, the cells were washed with PBS and fluorescence was measured with a Synergy H1 microplate reader at 548/574 nm. The cells were further incubated with Hoescht (Sigma-Aldrich, Saint Louis, MO, USA) at 5 µg/mL for an additional 30 min and the plate was read again at 355/546 nm. 

### 2.11. Statistical Analysis

Data are expressed as mean ± SEM. Statistical significance of the differences between donor groups was assessed by unpaired Student’s *t*-test or Mann–Whitney test depending on the normality of the data, which were analyzed by Shapiro–Wilk test. Two-way ANOVA was used to assess the significance of treatment, donor, and possible treatment x donor interaction effects. The Bonferroni post-hoc test was used for multiple comparisons within groups. The software used was GraphPad Prism 8.4.0 (GraphPad Software, Inc., Boston, MA, USA). The threshold of significance was set at *p* < 0.05.

## 3. Results

### 3.1. Oxidative Phosphorylation Is the Most Altered Pathway in LRMSC from IPF Patients

Transcriptome analyses were performed to identify the altered pathways in the LRMSC of IPF patients by comparing the gene expression to CON LRMSC. A heatmap illustrated a different expression profile between IPF and CON experimental groups (Figure 1A). In particular, 114 probes passed the filter criteria, 92% of which had a lower expression in IPF patient cells (Figure 1B). Genes with a significantly different expression included *mitochondrially encoded ATP synthase 6* (*MT-ATP6*), *mitochondrially encoded NADH dehydrogenases* (*MT-ND3*, *MT-ND4*, *MT-ND5*), *ribosomal proteins* (*RPL27a*, *RPS8*, *RPS2*), *eukaryotic translation initiation factors* (*eIF3a*, *eIF5*), *heat shock protein family A (Hsp70) member 8* (*HSPA8*), and *inosytol-3-phosphaten synthase 1* (*ISYNA1*).

ORA of significantly different expressed genes using KEGG revealed that the most significantly different pathway was oxidative phosphorylation, which was downregulated in IPF LRMSC (Figure 2A). Coinciding with ORA, GSEA also indicated that oxidative phosphorylation was the most different pathway between the IPF and CON groups, with normalized enrichment scores (NES) of −2 (*p* = 8.0 × 10^−6^) and −2.06 (*p* = 4.9 × 10^−6^) with the KEGG and Wikipathways databases, respectively. Thus, both ORA and GSEA results support that the oxidative phosphorylation pathway was compromised in IPF cells. Moreover, according to GSEA, IPF cells presented a downregulation of pathways related to oxidative stress and an upregulation of two pathways related to nutrient metabolism, such as unsaturated fatty acid biosynthesis and retinol metabolism (Figure 2B).

In agreement with pathway analysis, the study of clustered GO terms according to *p*-value revealed that IPF vs. CON differences in the processes related to aerobic electron transport chain were some of the most represented (Figure 3A). However, the analysis of clustered BP uncovered that the mesenchyme development, which includes epithelial to mesenchymal transition (EMT), was the BP with the highest representation. In addition, other BPs known to be altered in IPF patients were also found to be significantly different in LRMSC from IPF vs. CON, including cartilage homeostasis, complement activation, cell migration, collagen metabolism, and regulation of epithelial cell proliferation.

To identify the most significantly dysregulated BP in IPF LRMSC, the top up- and downregulated clustered BP were selected according to their normalized enrichment score (Figure 3B). Thus, considering the BP with an absolute NES higher or lower than 2, IPF LRMSC presented an enrichment in the expression of genes related to complement activation and mesenchyme development, whereas, on the other hand, aerobic electron transport chain was downregulated in IPF vs. CON LRMSC. 

All in all, the transcriptome analysis revealed a major downregulation of oxidative phosphorylation in IPF LRSMC and, therefore, we focused on the study of LRMSC mitochondria features.

### 3.2. LRMSC from IPF Patients Present an Impaired Mitochondrial Respiration Capacity Compared to Control LRMSC

To confirm that genes involved in the oxidative phosphorylation were downregulated in IPF LRMSC, we assessed the mRNA expression levels of COX-IV and ATP6 by real-time PCR in the samples from the cohorts used in the transcriptome analyses and from the Lund cohort. Indeed, the mRNA levels of both genes were significantly lower in IPF vs. CON LRMSC (Figure 4A). Further, a functional assay was conducted to elucidate whether such a downregulation has an impact on mitochondrial oxidative respiration. The oxygen consumption rate of IPF LRMSC was below that of CON LRMSC throughout the assay (Figure 4B). The functional assay analysis indicated that oxidative phosphorylation capacity was compromised in IPF LRMSC. Specifically, IPF cells exhibited a significantly lower basal and maximal respiration than CON LRMSC, as well as a lower spare respiratory capacity, which indicated a lower capability of mitochondria to respond to an energetic challenge (Figure 4C).

### 3.3. Mitochondrial Turnover Capacity Is Compromised in IPF LRMSC and Linked to Reduced Autophagy and Apoptosis

To ascertain whether the lower mitochondrial respiration exhibited by IPF LRMSC was related to the quantity of mitochondria, we assessed the mitochondrial-to-nuclear DNA ratio. No differences were found in the mitochondrial DNA quantification between IPF and CON LRMSC (Figure 5A). The reduced respiratory activity of IPF cells along with the maintenance of the number of mitochondria could indicate that mitochondrial dynamics were impaired in IPF LRMSC, as has been previously demonstrated in IPF lung epithelial cells [12], fibroblasts [31], and macrophages [32]. To test this hypothesis, the mRNA expression levels of the key genes involved in mitochondrial biogenesis, fusion, and removal were analyzed (Figure 5B). The expression of the key regulator of mitochondrial biogenesis, *PGC1A*, was low in LRMSC, being close to the detection limit in several IPF samples. On the other hand, the expression of *MFN2*, an essential gene for mitochondrial fusion, was significantly lower in IPF patients. Likewise, the expression of two key genes involved in the elimination of dysfunctional mitochondria, namely, *PINK1* and *PGAM5L*, was significantly lower in LRMSC from IPF patients. Altogether, these gene expression results suggest a mitochondrial recycling slowdown in IPF cells. The unaltered quantity of mitochondria in the face of a compromised capacity to recycle mitochondria entails the presence of less functional and damaged mitochondria. The impaired mitochondrial respiration in IPF cells was previously shown (Figure 4B,C). To evaluate mitochondrial damage, we analyzed the integrity of the mitochondrial membrane and the morphology of the mitochondria. No significant differences were found in the mitochondrial membrane integrity between IPF and CON cells as measured by TMRM assay (Figure 5C). However, the mitochondria observed in representative IPF samples by TEM were found to be irregularly shaped, having thickened membranes and non-defined crests, compared to cells from CON subjects (Figure 5D).

Next, we aimed to elucidate whether these mitochondrial dysfunction signs were linked to autophagy and apoptosis. The autophagy response was significantly decreased in IPF vs. CON LRMSC, as indicated by the number of autophagic vacuoles (Figure 5E). As is shown in Figure 5B, IPF LRMSC presented a lower *PGAM5L* mRNA expression, a mitochondrial protein that, in addition to its role as a PINK1 stabilizer [33], triggers caspase activation and cell death [34]. Moreover, the expression of other pro-apoptotic genes, *p53* and *CASP8*, was significantly lower in IPF compared to CON LRMSC (Figure 5F). Altogether, these data suggest that IPF LRSMC present a reduced capacity to execute mitophagic cell death.

### 3.4. IPF and CON LRMSC Exhibit a Different Mitochondrial Response to TGFβ Exposure

In human cell lines, TGFβ induces mitochondrial depolarization and a rapid increase in PINK1 expression in epithelial cells [34], whereas it alters mitophagy and promotes the occurrence of mitochondrial function defects in fibroblasts during myofibroblast differentiation [35]. In the present work, we sought to investigate the effect of TGFβ exposure on LRMSC at different times and concentrations. The 10 ng/mL TGFβ treatment for 24 h did not change mitochondria integrity or number in either IPF or CON LRMSC (Figure 6A,B, respectively). Under TGFβ conditions, both mitochondria integrity and number of IPF LRMSC diverged from CON LRMSC, although without reaching statistical significance. Next, we analyzed the response in the expression of genes related to oxidative phosphorylation and mitochondrial dynamics. In a first experiment, LRMSC were treated with different TGFβ concentrations for 24 h and, in the second experiment, with 10 ng/mL for a longer exposure.

Upon 24 h TGFβ exposure, *COX-IV* and *PGAM5L* expression responses were different depending on the donor, with IPF cells displaying a significantly lower expression for both genes (Figure 6C). However, TGFβ treatment did not elicit changes in the expression of mitochondria-related genes in either IPF or CON LRMSC. Upon 48 h TGFβ exposure (Figure 6D), IPF cells exhibited a significantly lower *COX-IV* and *PINK1* expression response and, within TGFβ-treated cells, IPF LRMSC presented significantly lower *COX-IV*, *MFN2*, and *PINK1* mRNA levels than CON cells. Moreover, TGFβ treatment for 48 h upregulated *COX-IV* and *ATP6* expression, but the induction in *ATP6* expression was only significant in CON cells.

## 4. Discussion and Conclusions

In this work, we demonstrated that IPF LRMSC present signs of mitochondrial dysfunction. The identification of molecular and cellular features in IPF lung tissue is necessary to understand IPF etiopathogenesis. There is a significant amount of literature describing the alterations of lung cells, particularly AEC, fibroblasts, and immune cells, in IPF. In contrast, works analyzing IPF LRMSC alterations are scarce [25,36], despite LRMSC exerting a pivotal role on tissue repair and regeneration. The unknown role of LRMSC in IPF prompted us to analyze the differential characteristics of IPF LRMSC using comparative transcriptomic and functional approaches.

Our transcriptomic results provided us with the first clues to understanding the behavior of IPF LRMSC and indicated that mitochondrial dysfunction was a major feature of IPF LRMSC. We further demonstrated that mitochondrial respiration declined and the capacity to generate, fuse, and remove mitochondria was compromised, resulting in morphologically altered mitochondria. Mitochondria dysfunction also occurred in IPF differentiated lung cells both in fibroblasts and in epithelial cells [12,31], in which they exerted a causative role on IPF progression. The occurrence of mitochondrial dysfunction in the cells from which lung fibroblasts and epithelial cells are derived may imply a significant impact on IPF.

The impaired respiration capacity of IPF LRMSC indicated that the metabolic shift described in the IPF lung was also present in lung precursor cells. Such an impairment, together with the metabolite alterations that suggest a reduced β-oxidation in the IPF lung [37], indicates that energy production from mitochondrial activity is less predominant in IPF cells, most probably coming from fibroblasts [38], alveolar macrophages [32], and LRMSC, as we have shown. The impaired respiration capacity could be explained not only by the reduced expression of mitochondrial chain components, but also by the decreased mitochondrial biogenesis and turnover capacity, which would further negatively impact the mitochondrial respiration efficiency of IPF LRMSC, as these organelles aged, according to our results. As Mora and others have previously demonstrated, the defective mitochondrial turnover is a crucial feature in IPF. The expression of *PGC1A*, *MFN2*, and *PINK1* was found to be repressed in differentiated IPF lung cells, and their inactivation induced pulmonary fibrosis in animal models [12,31,39]. PGAM5 is a protein involved in mitophagy and apoptosis [33,40] for which a role in experimental lung fibrosis has been described [41]. Our results suggest that *PGAM5L* downregulation may limit, together with *PINK1* downregulation, mitophagy in IPF LRMSC, as PGAM5 functions as a PINK1 stabilizer. This was also supported by the reduced ability to activate autophagy in our IPF cells, which is in line with the reduced autophagy flux characteristic of IPF fibroblasts [35]. Moreover, given the role of PGAM5L in apoptosis, its downregulation may contribute to the cell resistance to apoptosis, also suggested by the decreased expression of *p53* and *CASP8* genes in IPF LRMSC. In fibroblasts, mitochondrial dysfunction and a decrease in autophagy activity contribute to apoptosis resistance [38,42], which is further reinforced by the development of senescence [10]. Conversely, in healthy conditions, mitochondrial dysfunction and damage can trigger mitophagy and autophagy, whereas the inability to recover from mitochondrial damage promotes cell senescence or apoptosis [43]. However, apoptosis is not activated in IPF fibroblasts [4], which constitutes a key pathological mechanism of IPF that could be extended to LRMSC. Then, both LRMSC and fibroblasts would share the reduced autophagy and apoptosis interlinked responses, whereas, on the contrary, in epithelial cells apoptosis is known to be activated [44,45]. IPF LRMSC exhibiting an impaired autophagy and persistent survival would imply that they maintain their mitochondria despite being old and less functional. Presumably, cells arising from IPF LRMSC would inherit mitochondrial defects, thereby contributing to IPF mitochondrial dysfunction.

The response of genes related to oxidative phosphorylation and mitochondrial dynamics to TGFβ exposure was different between IPF and CON LRMSC. CON cells showed higher expression levels than IPF LRMSC under the presence of TGFβ, particularly evident in the prolonged exposure. These results suggest that IPF LRSMC may not activate mitochondrial respiration and turnover upon a profibrotic environment and are in accordance with the insufficient mitophagy occurring in myofibroblast differentiation [35,46]. This highlights the relevance of mitochondrial function regulation in lung fibrosis, as evidenced in recent works in which the restoring of mitochondrial function [47] and homeostasis [48] has been shown to reduce lung fibrosis.

On the other hand, our transcriptomic analysis also supports the hypothesis that LRMSC may be predisposed to differentiating into collagen-producing fibroblasts. BP related to mesenchyme development were the most abundant, and collagen biosynthesis was also significantly upregulated BP in IPF vs. CON subjects (Figure 3A,B). Then, as LRMPC are committed to differentiating into myofibroblasts [14,15], LRMSC might already be predisposed to follow this lineage [49]. In IPF LRMPC, fibrogenicity is conferred by S100 calcium-binding A4 protein, the epigenetic CD44/BRG1 nuclear complex, and other nuclear proteins involved in repair, stemness regulation, and cell viability [50,51,52] and mediated by IL-8 [53]. Moreover, LRMPC from IPF patients show a loss of transcripts encoding for DNA damage response and DNA repair components and promote the senescence of their progeny [54]. Further studies are required to understand the potential involvement of LRMSC in fibrogenicity and the underlying molecular mechanisms.

Other results obtained in the transcriptomic analysis agree with previous findings in IPF patients. For instance, we observed a downregulation of ribosomal protein genes (Figure 2A) in accordance with the recent finding in fibroblasts at an early stage of IPF [55] and suggesting a loss of proteostasis. The BP with the highest significant NES in IPF LRMSC is complement activation. The dysregulation of the innate immune system has been posed as a critical driver of IPF [56], and the dysregulation of those genes included in this pathway in non-immune cells may contribute to the proinflammatory response, which was previously demonstrated in a small cohort of IPF patients [25].

Moreover, we observed an upregulation of genes involved in amino acid metabolism in IPF LRMSC, particularly in glycine, serine, and threonine (Figure 3B). Accordingly, the upregulation of phosphoglycerate dehydrogenase, an enzyme involved in serine synthesis and collagen production, was previously found to be increased in IPF patients [57]. In addition, the levels of specific amino acids were found to be increased in lung tissue and exhaled air of IPF patients [58,59]. Changes in amino acid metabolism, together with the upregulation of genes involved in retinol and unsaturated fatty acid metabolism, points to particular metabolic changes occurring in LRMSC as the cell-type-specific metabolic reprogramming described in IPF differentiated lung cells [60]. Increased retinol metabolism might be related to the ability of vitamin A derivatives to promote cell plasticity and activate catabolism [61,62], which might be interpreted as an attempt to meet the elevated energy requirements characteristic of the IPF lung [63]. The increased capacity for unsaturated fatty acid biosynthesis found in IPF LRMSC agrees with the growing evidence indicating the crucial role of dysregulated lipid metabolism in lung fibrosis [37,64].

The small number of IPF patients included for LRMSC isolation is the main limitation of this study. Therefore, given the heterogeneity of IPF, the investigation of mitochondrial parameters in LRMSC from a larger number of patients with different phenotypes is needed to confirm the results presented here. The transcriptomic results were obtained using only four individuals in each group from the Spanish cohorts, but a validation by PCR and functional assays in samples from the Lund cohort was performed. Finally, the limited number of cells obtained per patient impeded the validation at the protein level of the expression of genes involved in mitochondrial dynamics.

This is one of the first studies to characterize IPF LRMSC and contributes to the knowledge of molecular and cellular mechanisms involved in the pathology of IPF. Future studies in LRMSC are needed to understand the impact of mitochondrial dysfunction on IPF onset and progression.

## Figures and Tables

**Figure 1 cells-12-02084-f001:**
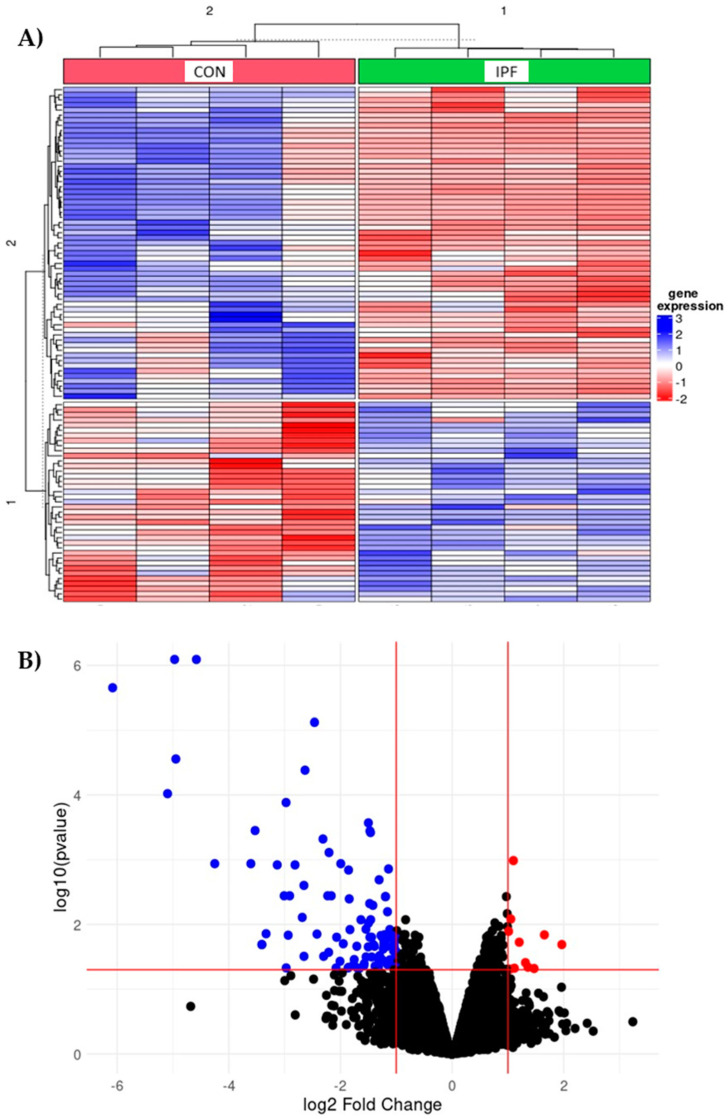
Differentially expressed genes in IPF vs. CON LRMSC. Genes with absolute fold change >2 and adjusted *p*-value for false discovery rate (FDR) <0.05 were considered significant. (**A**) Heatmap with top 100 differentially expressed genes. (**B**) Volcano plot highlighting the upregulated (in red) and downregulated (blue) genes with an absolute fold change ≥2 and an adjusted *p*-value <0.05.

**Figure 2 cells-12-02084-f002:**
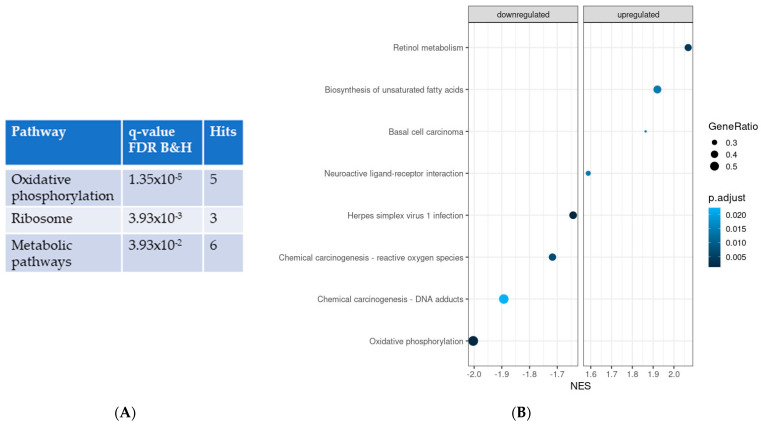
Significantly different pathways in IPF vs. CON LRMSC. Pathways were identified using the Kyoto Encyclopedia of genes and genomes by applying an adjusted *p*-value for B&H <0.05. (**A**) Over-represented analysis. (**B**) Gene-set enrichment analysis. NES: normalized enrichment score.

**Figure 3 cells-12-02084-f003:**
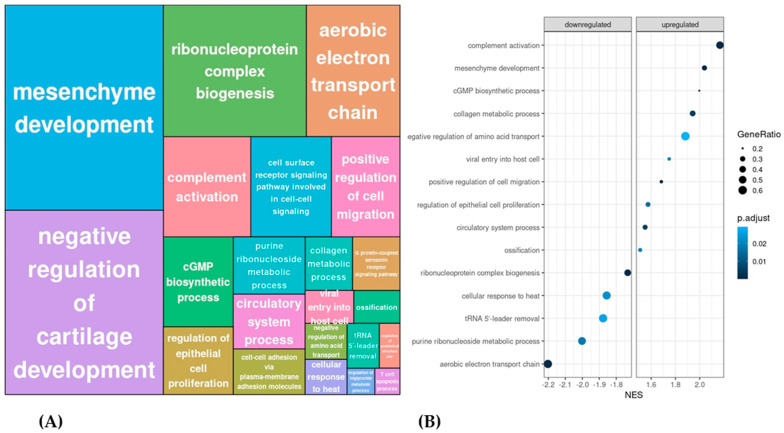
Significantly different biological processes in IPF vs. CON LRMSC. (**A**) Related biological processes with a 0.8 correlation were clustered according to *p*-value. (**B**) The top 10 up- and downregulated biological processes were selected according to their normalized enrichment score (NES).

**Figure 4 cells-12-02084-f004:**
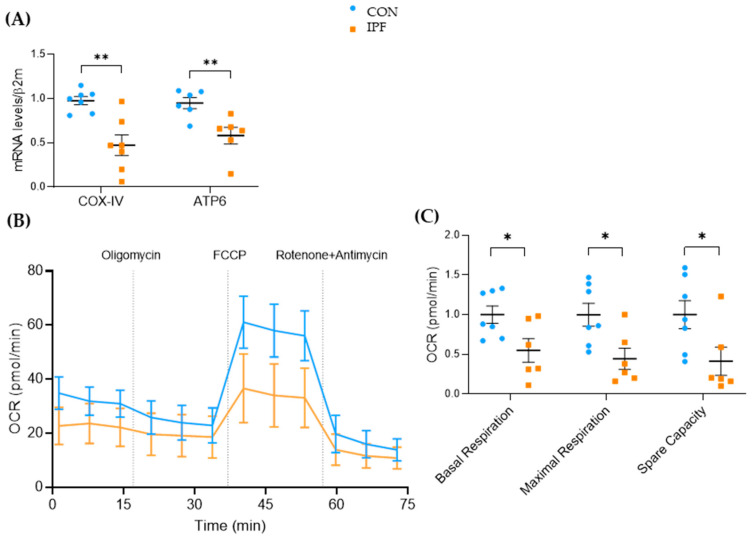
Mitochondrial respiration capacity is lower in LRMSC from IPF patients. (**A**) COX-IV and ATP6 mRNA levels assessed by real-time PCR. (**B**) Real-time oxygen consumption rate (OCR) analyzed by Seahorse XF Cell Mito Stress Test. (**C**) Basal respiration, maximal respiration, and spare capacity. N = 6–7. Mean ± SEM, non-parametric Student’s *t*-test, * *p* < 0.05, ** *p* < 0.01.

**Figure 5 cells-12-02084-f005:**
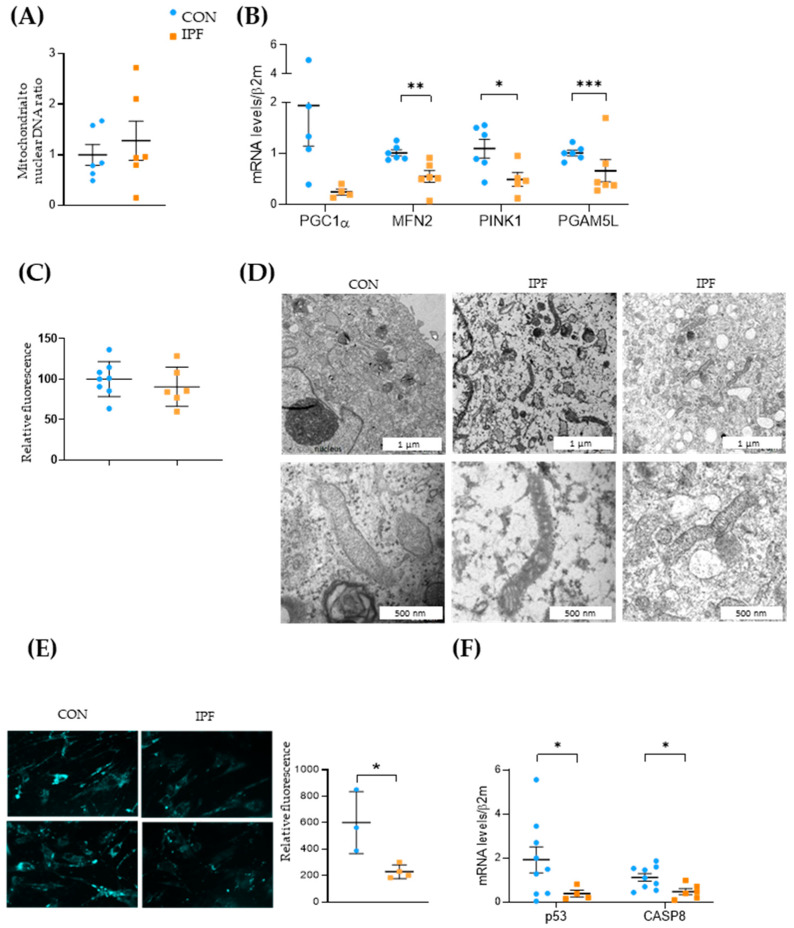
Mitochondrial dynamics is impaired in IPF LRMSC. (**A**) Mitochondrial and nuclear DNA assessed by real-time PCR. (**B**) mRNA expression of genes involved in mitochondrial biogenesis (*PGC1A*), fusion (*MFN2*), and mitophagy (*PINK1*, *PGAM5L*). CON. (**C**) Mitochondrial membrane integrity measured by TMRM assay. CON. (**D**) Transmission electron microscopy (TEM) images of representative LRMSC. Arrows indicating mitochondria. (**E**) Detection of autophagic vacuoles by fluorescence (left panel) and quantification of relative fluorescent units (right panel). (**F**) mRNA expression of genes involved in apoptosis. Mean ± SEM. Non-parametric Student’s *t*-test for all experiments, * *p* < 0.05, ** *p* < 0.01, *** *p* < 0.005.

**Figure 6 cells-12-02084-f006:**
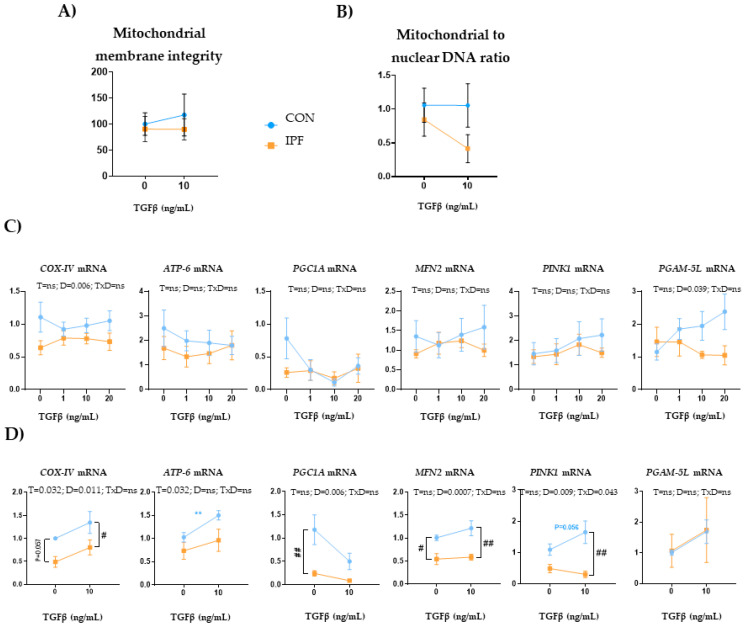
TGFβ exposure triggers different mitochondrial responses in IPF and CON LRMSC. Cells were treated with vehicle or TGFβ at the indicated concentration for 24 h (**A**–**C**) or 48 h (**D**). (**A**) Mitochondrial and nuclear DNA quantification assessed by PCR. (**B**) Fluorescence quantification in the TMRM assay. (**C**) mRNA levels of mitochondrial genes. N = 6–9. Mean ± SEM two-way ANOVA, T: treatment effect; D: donor effect, Bonferroni post-hoc. # CON vs. IPF, # *p* < 0.05, ** and ## *p* < 0.01.

**Table 1 cells-12-02084-t001:** Patient information. Gender and age of the IPF and CON patients recruited in this study. IPF: idiopathic pulmonary fibrosis patients; CON: control patients.

Number	Experimental Group	Cohort	Gender	Age
1	IPF	Bellvitge	m	65
2	IPF	Bellvitge	f	68
3	IPF	Bellvitge	m	74
4	IPF	Bellvitge	m	68
5	IPF	Bellvitge	m	62
6	IPF	Bellvitge	m	70
7	IPF	Bellvitge	m	66
8	IPF	Lund	m	68
9	IPF	Lund	f	62
10	IPF	Lund	m	67
11	IPF	Lund	f	65
12	CON	HUSE	f	39
13	CON	HUSE	m	77
14	CON	HUSE	m	75
15	CON	HUSE	m	62
16	CON	HUSE	m	65
17	CON	HUSE	m	70
18	CON	HUSE	m	57
19	CON	HUSE	m	77
20	CON	Lund	m	68
21	CON	Lund	m	43
22	CON	Lund	m	62
23	CON	Lund	m	39
24	CON	Lund	f	44
25	CON	Lund	m	62

## Data Availability

Data are available on request.

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
