# Peer review of "Mitochondrial Dysfunction in Lung Resident Mesenchymal Stem Cells from Idiopathic Pulmonary Fibrosis Patients"

_cells, 2023, doi:10.3390/cells12162084_

Round 1

Reviewer 1 Report

This is an interesting and important study. The contents described in this MS fall into the scope of journal. This study was well designed and presented. The results are sound and scientifically interpreted. I don't have major concern about this study. However, any validation data at protein level by immunoblotting, IF or IHC that can be presented in Fig 5 and 6 will strength the study. At least, the authors better to mention the limitation of validation at protein level in discussion.

Author Response

Response to reviewer 1

We thank the reviewer for the comments. We agree with the reviewer in that the analysis of protein levels in Figures 5 and 6 will strengthen these results. Unfortunately, not only the number of patients was limited, but also the number of cells isolated from a patient. This means that experiments were carefully designed considering the available number. Moreover, the experiments included in Figs 5 and 6 were performed at the end of the study, when an important part of the cells had been already used. Since protein analysis requires a higher number of cells than mRNA analysis, we decided to analyse gene expression by PCR and prioritized electromicroscopy analysis and the autophagy functional assay that do not require a very high number of cells. As suggested by the reviewer, we mentioned the limitation in the discussion section (lines 557-559).

Reviewer 2 Report

In this study, the authors made transcriptomic and functional analyses of lung resident mesenchymal stem cells from IPF- patients (IPF-LRMSC) versus individuals without interstitial lung disease as controls (CON-LRMSC). The aim of the study is clear, the experimental design is adequate and the results are clear and compelling.

With the intent to give completeness work done by the authors, I have some suggestions that should be incorporated into the manuscript  

1.     In the introduction: A recent and interesting review, Ligresti G, Raslan AA, Hong J, Caporarello N, Confalonieri M, Huang SK. Mesenchymal cells in the Lung: Evolving concepts and their role in fibrosis Gene. 2023 Apr 5;859:147142. Doi:10.1016/j.gene.2022.147142. Epub 2023 Jan 2 PMID: 36603696

2.     In the results section: The design of the experimental array of mRNA expression data available in Gene Expression Omnibus

3.     And in the discussion include:

Wang C, Cao H, Gu S, Shi C, Chen X, Han X. Expression analysis of microRNAs and mRNAs in myofibroblast differentiation of lung resident mesenchymal stem cells. Differentiation 112: 10–16, 2020. doi:10.1016/j.diff.2019.11.002.

Adams, T.S., Schupp, J.C., Poli, S., Ayaub, E.A., Neumark, N., Ahangari, F., et al., 2020. Single-cell RNA-seq reveals ectopic and aberrant lung-resident cell populations in idiopathic pulmonary fibrosis. Sci. Adv. 6 (28), eaba1983.

Author Response

Response to Reviewer 2

  1. We also think that the review recently published by Ligresti G and colleagues is very interesting. We have incorporated this cite (number 4) in the introduction section twice (lines 54 and 57) and in the discussion section when we discuss about fibroblast persistence to apoptosis (line 502). We thank the reviewer for this suggestion.
  2. We thank the reviewer for pointing this out. Accordingly, we have sent to GEO the microarray information, which includes experimental design, protocols, technical details and probeset data. This information will be available if the present manuscript is accepted for publication from September the 1st. The GEO confirmation mail is attached below.
  3. The paper published by Wang C et al describes the expression profile in differentiating LRMSC into myofibroblast. In the revised manuscript, we used this cite (number 49) to support the idea that LRMSC might be predisposed to differentiate into myofibroblasts (line 523). We think that the paper published by Adams TS et al is pioneering research using single-cell RNA-seq technology in IPF. However, this work does not directly deal with LRMSC. In our manuscript, the results and conclusion of Adams’ paper were not used in any sentence or rationale and, therefore, we think that is not appropriate to include this cite. We thank the reviewer for these suggestions.
